# DCLK1 Drives EGFR-TKI-Acquired Resistance in Lung Adenocarcinoma by Remodeling the Epithelial–Mesenchymal Transition Status

**DOI:** 10.3390/biomedicines11051490

**Published:** 2023-05-22

**Authors:** Rui Yan, Xuying Huang, Heshu Liu, Zeru Xiao, Jian Liu, Guangyu An, Yang Ge

**Affiliations:** Beijing Chao-Yang Hospital Department of Oncology, Capital Medical University, 8 Gongren Tiyuchang Nanlu Road, Chaoyang Dist., Beijing 100020, China

**Keywords:** lung adenocarcinoma, EGFR-TKI resistance, doublecortin-like kinase 1, epithelial-mesenchymal transition

## Abstract

Objective: Epidermal growth factor receptor–tyrosine kinase inhibitor (EGFR-TKI) is a first-line treatment for lung adenocarcinoma with EGFR-sensitive mutations, but acquired resistance to EGFR-TKIs remains a problem in clinical practice. The development of epithelial–mesenchymal transition (EMT) is a critical mechanism that induces acquired resistance to TKIs. Reversing acquired resistance to EGFR-TKIs through targeting the key molecules driving EMT provides an alternative choice for patients. We, therefore, aimed to explore the role of doublecortin-like kinase 1 (DCLK1) as an EMT driver gene in the acquired resistance of lung adenocarcinoma to EGFR-TKIs. Methods: The IC_50_ of Gefitinib or Osimertinib in PC9/HCC827 cells was measured using a cell counting kit-8 (CCK8) assay. The expression levels of EMT-related genes in PC9 and HCC827 cells were detected using RT-PCR and Western blot. Cell migration and invasion abilities were assessed via a transwell assay. For the in vivo experiments, PC9 cells were subcutaneously injected into BALB/c nude mice to form tumors. Upon harvesting, tumor tissues were retained for RT-PCR, Western blot, and polychromatic fluorescence staining to detect biomarker changes in the EMT process. Results: Gefitinib-resistant PC9 (PC9/GR) and Osimertinib-resistant HCC827 (HCC827/OR) cells showed remarkable activation of EMT and enhanced migration and invasion abilities compared to TKI-sensitive cells. In addition, DCLK1 expression was markedly increased in EGFR-TKI-resistant lung adenocarcinoma cells. The targeted knockout of DCLK1 effectively reversed the EMT phenotype in TKI-resistant cells and improved EGFR-TKI sensitivity, which was further validated by the in vivo experiments. Conclusions: DCLK1 facilitates acquired resistance to EGFR-TKI in lung adenocarcinoma by inducting EMT and accelerating the migration and invasion abilities of TKI-resistant cells.

## 1. Introduction

Lung cancer is a pressing global health concern, with non-small cell lung cancer (NSCLC) being the most prevalent subtype, representing approximately 85% of all cases. Within the spectrum of NSCLC subtypes, lung adenocarcinoma warrants urgent attention due to its high incidence rates [1,2]. In the Asian population, 50–60% of lung adenocarcinoma patients have EGFR-sensitive mutations (the exon 19 deletion mutation and the exon 21 L858R point mutation) [3]. For these patients, it is consistently recommended in current guidelines to use epidermal growth factor receptor–tyrosine kinase inhibitors (EGFR-TKIs) as the preferred treatment [4,5,6,7]. Although the objective response rate (ORR) of initial EGFR-TKI therapy is impressive, acquired resistance is inevitably developed in a significant proportion of patients [8,9]. One crucial reason for the development of EGFR-independent TKI resistance is the occurrence of epithelial–mesenchymal transition (EMT) [9,10], but the underlying mechanism of EMT has not been fully elucidated.

EMT, which describes the conversion of polarized epithelial cells into motile mesenchymal cells, is a normal physiological process of embryonic development. However, abnormal activation of EMT leads to various pathological conditions, notably tumorigenesis and tumor progression [1,11]. Recent studies have shown that excessive activation of EMT can lead to acquired resistance of lung adenocarcinoma cells to EGFR-TKI [12]. Multiple lines of evidence have confirmed that TKI-resistant lung adenocarcinoma cells undergo altered mesenchymal morphology accompanied by the downregulation of epithelial markers and upregulation of mesenchymal markers, indicating the emergence of the EMT process [12,13]. In addition, many researchers suggest that the EMT process occurs independently or concomitantly with other resistance mechanisms, such as mutations in the EGFR-T790M locus or *MET* gene amplification, in patients with acquired resistance to EGFR-TKI [13]. Although targeting mesenchymal-resistant cells has been suggested in previous studies as a promising strategy for attacking acquired resistance to EGFR-TKI, it remains unclear whether we can reverse the onset of EMT by targeting driver molecules. Therefore, defining the driving genes of the EMT process may provide opportunities for follow-up clinical therapeutics of EGFR-TKI-resistant patients.

Doublecortin-like kinase 1 (DCLK1) is a microtubule-associated protein initially identified in the nervous system and has been shown to be involved in neurogenesis and microtubule migration [14,15]. A previous crystallography study revealed that DCLK1 contains multiple functional structural domains. The N-terminal of DCLK1 has two tandem doublecortin domains (DCX1: aa54-152 and DCX2: aa180-263), which are mainly responsible for binding to microtubules, and the C-terminal harbors a serine/threonine kinase domain, which is highly Ca^2+^-dependent (KD: aa374-648). The distinctive feature domain connecting the DCX domains to the kinase region, named the PEST region, consists of a linker of about 100 amino acids [16,17,18]. Evidence suggests that DCLK1 plays a crucial role in tumor development beyond its essential function in the nervous system. Several studies have also revealed that DCLK1 is highly expressed in multiple tumors and is recognized as a putative marker in certain tumor stem cells. DCLK1 is also involved in the initiation of gastrointestinal (GI) tumors, such as intestinal cancer, pancreatic cancer, and cholangiocarcinoma [19,20,21]. In addition, DCLK1 plays a crucial role in activating EMT by regulating the expression of EMT-associated genes. It upregulates mesenchymal markers, such as Snail, N-cadherin, and ZEB1, while downregulating epithelial markers, such as ZO1 and E-cadherin [22,23].

Recently, DCLK1 has been increasingly studied for its role in lung cancer. Studies have shown that an upregulation of DCLK1 suggests poor prognosis in lung cancer patients [24,25]. Panneerselvam et al. found that DCLK1 participates in the activation of EMT in lung cancer, and high expression of DCLK1 promotes the development of cisplatin resistance [26]. In addition, our previous study confirmed that the increased expression of DCLK1 in lung adenocarcinoma contributes to EGFR-TKI-acquired resistance, partly due to the maintenance of tumor cell stemness [27]. However, it is still unknown whether DCLK1 plays a role in EGFR-TKI-acquired resistance through the regulation of EMT activation.

In this study, the biological function of DCLK1 in EGFR-TKI-resistant lung adenocarcinoma via remodeling the EMT process was explored through in vitro and in vivo experiments, with the aim of finding potential therapeutic targets in EGFR-TKI-resistant lung adenocarcinoma. Compared to sensitive cells, we found significant EMT activation in Gefitinib-resistant (PC9/GR) and Osimertinib-resistant (HCC827/OR) cells. Additionally, DCLK1 expression was elevated in EGFR-TKI-resistant cells through remodeling the EMT process. In contrast, inhibition of DCLK1 reversed the occurrence of EMT and reduced the resistance of lung adenocarcinoma cells to EGFR-TKI. Collectively, it is confirmed through both in vitro and in vivo experiments that DCLK1, as a driving gene for EMT activation, contributes to EGFR-TKI resistance through remodeling the EMT program.

## 2. Materials and Methods

### 2.1. Cell Culture

EGFR-TKI-sensitive lung adenocarcinoma cells, PC9 and HCC827, were obtained from Shanghai FuHeng biological technology Co., Ltd. (Shanghai, China) and American Type Culture Collection (ATCC), respectively. Both are epithelial adherent cells of lung adenocarcinoma with the EGFR exon 19 deletion mutation. PC9 Gefitinib-resistant cells, defined as PC9/GR, were purchased from Shanghai FuHeng Biological Technology Co., Ltd. and have a Gefitinib-resistance concentration of 2.2 μM. The HCC827 Osimertinib-resistant cell line, defined as HCC827/OR, was provided by Beijing Chest Hospital and has an Osimertinib-resistance concentration of 0.16 μM. PC9 and PC9/GR cells were cultured in Dulbecco’s modified Eagle’s medium (DMEM), while HCC827 and HCC827/OR cells were cultured in the RPMI-1640 medium. The basic medium was supplemented with 10% fetal bovine serum (FBS) and 1% penicillin/streptomycin (P/S) to create a complete medium. The cells were incubated under standard conditions of 37 °C and 5% CO_2_.

### 2.2. Transwell Assays

For the migration assay, 5 × 10^4^ cells were inoculated in the upper chamber of the Transwell without FBS, and 600 μL of complete medium was placed in the lower chamber. At 24 h later, the chambers were fixed with 4% paraformaldehyde, followed by 0.1% crystalline violet staining. The invasion assay was performed similarly to the migration assay, with additional Matrigel loaded to the upper chamber of the Transwell. The migrated and invaded cells were observed under a microscope and photographed, and the results were analyzed using the Image J software. The measurement parameters of Image J are as follows: Image–type: 8-bit; Image–adjust–threshold: 0/128; process–binary–watershed; and analyze–analyze particles: 0.01.

### 2.3. CRISPR-Cas9-Mediated Knockout of DCLK1

The single-guide RNA (sgRNA) targeting DCLK1 was designed according to the principle of the CRISPR/Cas9 system. The sgRNA-Oligos were annealed and ligated to a lentiviral vector (LentiCRISPR-V2). After the plasmid construction, it was mixed with lentiviral packaging plasmid psPAX2, PMD2G, and transfected 293FT cells using liposome for Lentivirus production. At 72 h later, the target cells were screened with 2 μg/mL of puromycin. The sgRNA sequence is as previously described [27]. 

### 2.4. DCLK1 Rescue in Knockout Cell Lines

The coding sequence region of DCLK1 targeted by sgRNA was synonymously mutated. The original sequence TGGTAGTCAGCTCTCTACTCC was substituted into AGGAAGCCAACTATCAACACC, which resulted in no change in the amino acid. Then, the sequence-modified DCLK1 was stably transfected into DCLK1-knockout cell lines using a lentiviral vector. The complement expression of DCLK1 was validated using Western blot.

### 2.5. IC_50_ Determination

The cells were inoculated in a 96-well plate at 5 × 10^3^/well and treated with Gefitinib and Osimertinib in gradient concentrations. At 72 h later, the original medium was removed; then, a medium containing 10% CCK8 was added and the mixture was incubated for 30~60 mins, followed by determination of the OD value on a microplate reader, and the IC_50_ value was calculated using the GraphPad software. The gradient concentrations of Gefitinib and Osimertinib used were as follows: 

Gefitinib:

PC9: 0, 0.002, 0.005, 0.008, 0.01, 0.02, 0.05, and 0.1 (μM). 

PC9/GR: 0, 0.125, 0.25, 0.5, 1, 2, 3, 4, 5, and 6 (μM). 

PC9/GR CTRL: 0, 0.5, 1, 2, 3, 4, 5, 6, and 8 (μM).

PC9/GR DCLK1-KO: 0, 0.25, 0.5, 1, 2, and 3 (μM).

PC9/GR DCLK1^−/−^ CTRL: 0, 0.25, 0.5, 1, 2, and 3 (μM). 

PC9/GR DCLK1^−/−^ Res: 0, 0.25, 0.5, 1, 2, 3, 4, and 5 (μM). 

Osimertinib:

HCC827: 0, 0.001, 0.01, 0.02, 0.06, 0.1, 0.2, and 0.4 (μM). 

HCC827/OR: 0, 0.1 0.15, 0.5, 1, 1.5, and 3.5 (μM). 

HCC827/OR CTRL: 0, 0.15, 0.5, 1, 1.25, 1.5, 1.75, 2, 3, and 4 (μM).

HCC827/OR DCLK1-KO: 0, 0.15, 0.5, 1, 1.25, 1.5, 1.75, 2, 3, and 4 (μM). 

HCC827/OR DCLK1^−/−^ CTRL: 0, 0.15, 0.5, 1.5, 3, 3.5, and 4 (μM).

HCC827/OR DCLK1^−/−^ Res: 0, 0.15, 0.5, 1, 1.25, 1.5, 1.75, 2, 3, and 4 (μM). 

### 2.6. Western Blotting and Antibodies 

Two kinds of lung adenocarcinoma cell lines, PC9 and HCC827, were cultured in 10 cm^2^ Petri dishes for 48 h, and then proteins were harvested; the number of cells was approximately 2 × 10^6^. Afterward, 1% protease inhibitor and 1% Phenylmethanesulfonyl fluoride (PMSF) were added to the RIPA lysis buffer. Whole cells were lysed with the RIPA buffer on ice, and protein quantification was performed using the BCA Protein Assay Reagent (Thermo Fischer, USA), followed by protein denaturation at 95 °C for 10 mins. The denatured proteins were separated by electrophoresis in 10% SDS-PAGE and then transferred to PVDF membranes that had been activated by methanol in equal amounts. The PVDF membranes were blocked with 8% nonfat milk and incubated with primary antibodies overnight at 4 °C. The primary antibodies used were as follows: anti-ZO1 (1:1000, 13663S, CST), anti-E-cadherin (1:1000, 14472S, CST), anti-N-cadherin (1:1000, 13116S, CST), anti-ZEB1 (1:1000, 70512S, CST), anti-Vimentin (1:1000, 5741S, CST), anti-Snail (1:1000, 3879S, CST), anti-DCLK1 (1:1000, 31704, abcam), and anti-β-actin (1:1000, AF5001, Beyotime). After washing the PVDF membranes with 1 × TBST buffer three times, they were incubated with specific HRP-conjugated secondary antibodies (1:8000, A0208, Beyotime) for 60 min at room temperature. Finally, the immunoreactive signals were visualized using ECL reagents (Millipore, USA). All immunoreactive signals were measured using the Image J software, with the control signals normalized and presented as a percentage of the control group. All grayscale analysis results are presented in this manuscript.

### 2.7. RNA Extraction and Real-Time PCR

Two kinds of lung adenocarcinoma cell lines, PC9 and HCC827, were cultured in 6 cm^2^ Petri dishes for 48 h, and then RNA was harvested; the number of cells was approximately 5 × 10^5^. Total RNA was extracted from the cells using TRIzol reagent according to the manufacturer’s instructions. cDNA was obtained through reverse transcription of 1 μg of RNA in a two-step process using the Hifair^®^ Ⅲ 1st Strand cDNA Synthesis SuperMix kit (Yeasen, China, 11141ES60). The quantitative real-time PCR was performed with Hieff^®^ qPCR SYBR Green Master Mix (Low Rox Plus), and the amplification conditions were 95 °C for 5 min, 95 °C for 10 s, followed by 60 °C for 30 s. The number of cycles was 40 and the procedure was run on the 7500 Sequence Detection System (Applied Biosystems), using GAPDH as an internal reference and calculating the relative mRNA alterations based on 2^−ΔΔCt^. All primer sequences used in this research are listed in Appendix A.

### 2.8. Xenograft Tumor Model in Nude Mice

SPF-conditioned female BALB/c nude mice that were 5–6 weeks old were randomly divided into 4 groups (5 mice/group). PC9, PC9/GR, PC9/GR DCLK1-KO, and PC9/GR DCLK1-Rescue cells were inoculated subcutaneously, and the number of cells was 2 × 10^6^ per mouse. At 21 days later, the mice were sacrificed via CO_2_ inhalation, and tumor tissues were subsequently harvested. For the histopathological analysis, the tissue samples were taken for fixation, embedding, and sectioning. For the biochemical analysis, the samples were taken for protein and RNA extraction as above. A 5–6 mm^3^ mice tumor tissue was used for fixation in 4% paraformaldehyde for about 48 h. The volume ratio of tumor tissue to 4% paraformaldehyde was 1:7. A 2–3 mm^3^ mice tumor tissue was used for RNA and protein extraction. The animal experiment of this study was approved by the animal ethics committee of the Capital Medical University (Ethics Number: AEEI-2021-300).

### 2.9. Multiple Immunofluorescence Staining

Formalin paraffin-embedded tissue was derived from mouse subcutaneous tumor tissue. The paraffin-embedded tissue was sectioned continuously to 5 μm thickness for multiple immunofluorescence staining using Opal™4 Multiplex reagent (PerkinElmer, USA) according to the manufacturer’s specification. The primary antibodies used for immunofluorescence staining were E-cadherin (1:300, 14472S, CST) opal 690 and Vimentin (1:200, 5741S, CST) opal 570. Briefly, the slides were deparaffinized with xylene, rehydrated using a graded ethanol series, and fixed in 10% neutral-buffered formalin. The antigens were then repaired via microwave incubation with a citric acid antigen repair solution. After blocking for 20 min at room temperature, two consecutive staining procedures were performed, including incubation sequentially with a primary antibody (room temperature, 1 h), a horseradish peroxidase-conjugated secondary antibody (room temperature, 10 min), and an Opal fluorophore (room temperature, 5 min). Finally, the nuclei were counterstained with DAPI. The three-color Opal slides were visualized using the Vectra quantitative pathology imaging system (PerkinElmer). For all multicolor fluorescence images, each channel was obtained separately to further evaluate the gene expression levels according to the mean fluorescence intensity of the whole slide via the Image J software 1.8.0.

### 2.10. Data and Statistical Analysis

All data were analyzed using the GraphPad Prism 9.0 software and displayed as mean ± SEM. For the parametric analyses, statistical differences were determined using Student’s *t*-test. *p* < 0.05 was considered statistically significant, with * *p* < 0.05; ** *p* < 0.01; *** *p* < 0.001, and **** *p* < 0.0001.

## 3. Results

### 3.1. EGFR-TKI-Resistant Cells Have More Robust Migration and Invasion Abilities Than Sensitive Cells

First, we analyzed the IC_50_ of Gefitinib using the CCK8 assay and found that the IC_50_ of Gefitinib for PC9 cells was 15.99 nM, while the IC_50_ for PC9-resistant cells was 4.427 μM, which confirms the successful establishment of a Gefitinib-resistant cell line (Figure 1A). Considering that the activation of the EMT program is one of the crucial mechanisms of EGFR-TKI resistance, we evaluated the activation status of the EMT program in resistant cells as well. As expected, we found that compared to PC9-sensitive cells, PC9 Gefitinib-resistant cells were more likely to undergo EMT, showing decreased expression of *epithelial markers,* such as ZO1, and up-regulation of *mesenchymal markers,* such as ZEB1, Snail, and Vimentin (Figure 1B,C). Next, we examined the IC_50_ of HCC827 and HCC827/OR, which was 65.64 nM and 2.211 μM, respectively (Figure 1D). Similar to the results regarding PC9 cells, HCC827 Osimertinib-resistant cells also showed significant activation of EMT signature genes compared to sensitive cells, showing downregulation of E-cadherin expression accompanied by upregulation of mes*enchymal markers*, such as ZEB1, N-cadherin, and Vimentin (Figure 1E,F). In addition, by comparing the migration and invasion ability of TKI-sensitive and resistant cells via the transwell assay, we found that the malignant behavior of TKI-resistant lung adenocarcinoma cells, PC9/GR and HCC827/OR, was notably enhanced (Figure 1G,H). These results demonstrate that activation of the EMT program is accompanied by the occurrence of Gefitinib and Osimertinib resistance, a process that promotes the aggressiveness of EGFR-TKI-resistant cells.

### 3.2. Knockout of DCLK1 Suppresses the EMT Program of TKI-Resistant Cells

Consistent with a previous study [27], we verified that the expression of DCLK1 was robustly increased in PC9/GR cells compared to PC9 cells (Figure 2A). To further confirm whether DCLK1 mediates EGFR-TKI-acquired resistance, we knocked out DCLK1 in PC9/GR cells through CRISPR-Cas9 and verified successful DCLK1 depletion via Western blot (Figure 2B). Subsequently, we found the IC_50_ of Gefitinib was 4.118 μM and 1.924 μM for PC9/GR CTRL and PC9/GR DCLK1-KO cells, respectively, suggesting that the knockout of DCLK1 in PC9/GR cells leads to increased sensitivity to Gefitinib by about 2-fold (Figure 2C). Moreover, HCC827/OR cells also showed a significant elevation in DCLK1 expression (Figure 2D), while the knockout of DCLK1 restored the sensitivity of HCC827/OR cells to Osimertinib to a moderate extent (Figure 2E,F). To identify the role of DCLK1 in the EMT process, we investigated the effect of DCLK1 on EMT-related gene expression in TKI-resistant cells using RT-PCR and Western blot and found that DCLK1 knockout directly upregulated the *epithelial markers* ZO1 and E-cadherin, and conversely downregulated the *mesenchymal markers* ZEB1, Snail, Vimentin, and N-cadherin, suggesting a suppression of the EMT program (Figure 2G,H). In line with EMT activation, silencing of DCLK1 was accompanied by a weakening of the invasion and migration ability of TKI-resistant cells (Figure 2I,J). The above findings suggest that DCLK1, as a driver gene for EMT activation, mediates acquired resistance of lung carcinoma cells to Gefitinib and Osimertinib through remodeling the EMT process.

### 3.3. DCLK1 Rescue Restores the Malignant Phenotype of TKI-Resistant Cells

Next, we rescued the DCLK1 expression in DCLK1-knockout PC9/GR and HCC827/OR cells and verified DCLK1 expression in the DCLK1-rescued cell line via Western blot (Figure 3A,C). Subsequently, we tested the IC_50_ of DCLK1 knockout-resistant cells versus DCLK1-rescued cells using the CCK8 assay. As shown in Figure 3B, DCLK1-rescued PC9/GR cells (PC9/GR DCLK1^−/−^ Res) have an IC_50_ value of 3.577 μM, which is almost two times higher than that of PC9/GR DCLK1-KO cells (PC9/GR DCLK1^−/−^ CTRL) (1.786 μM). Consistently, the reintroduction of DCLK1 in HCC827/OR DCLK1-KO cells also reduces its sensitivity to Osimertinib (Figure 3D). To confirm whether DCLK1 plays a dominant role in EMT-induced acquired resistance to EGFR-TKI, we found that rescuing of DCLK1 reactivates the EMT process at both the mRNA and protein levels, with reduced expression of *epithelial markers* and enhanced expression of *mesenchymal markers* (Figure 3E,F). Additionally, the transwell assay also confirmed that DCLK1 rescue enhances the migration and invasion abilities of lung adenocarcinoma cells (Figure 3G,H). These results suggest that DCLK1 is a crucial molecule in EMT activation to obtain acquired resistance to EGFR-TKI.

### 3.4. DCLK1 Mediates EMT Program Activation in TKI-Resistant Cells In Vivo

To validate the above findings in vivo, we inoculated PC9, PC9/GR, PC9/GR DCLK1-KO (PC9/GR DCLK1^−/−^ CTRL), and DCLK1-rescued PC9/GR cells (PC9/GR DCLK1^−/−^ Res) subcutaneously in the abdomen of the mice, with 2 × 10^6^ tumor cells per mouse. After 21 days, we harvested tumor tissues for RNA and protein extraction. The RT-PCR results showed the EMT program was significantly activated in the Gefitinib-resistant tumors, whereas knockout of DCLK1 reversed the EMT process; however, when DCLK1 was re-expressed, it reinduced the expression of EMT-related genes, such as the downregulation of E-cadherin and upregulation of Vimentin and N-cadherin (Figure 4A). The Western blot results also showed that compared to the PC9 group, the *epithelial marker* E-cadherin was significantly downregulated in PC9/GR tumors. Moreover, the *mesenchymal markers* Vimentin and N-cadherin were upregulated, suggesting activation of the EMT program. In contrast, the EMT process was suppressed in the DCLK1^−/−^ PC9/GR cells, and it was triggered again after DCLK1 was rescued (Figure 4B). In addition, similar results were obtained from the immunofluorescence assay, confirming the involvement of DCLK1 in remodeling the EMT process and influencing the sensitivity of lung adenocarcinoma cells to EGFR-TKI (Figure 4C). 

In conclusion, both in vitro and in vivo experiments confirmed the vital role of DCLK1 in remodeling the EMT program to induce EGFR-TKI-acquired resistance in lung adenocarcinoma, suggesting that DCLK1 is an important regulatory gene that induces the activation of the EMT program, and DCLK1 is a molecule worthy of attention in the event of EGFR-TKI resistance in lung adenocarcinoma.

## 4. Discussion

Aberrant activation of EGFR downstream signaling pathways, mainly including PI3K/AKT/mTOR and MAPK/ERK, is a crucial trigger for the initiation of lung adenocarcinoma. Thus, targeted intervention of EGFR molecules on the cell surface with tyrosine kinase inhibitor (TKI) is the first-line treatment for patients with EGFR-sensitizing mutations [28,29,30]. However, despite the initial exciting efficacy, EGFR-TKI targeted therapy struggles to achieve long-lasting benefits since acquired resistance to EGFR-TKI will inevitably occur [31]. However, EGFR-T790M mutation, the most pivotal cause of first-generation TKI resistance, accounts for about 50% of the resistant population [32]. Third-generation TKIs can solve the problem of resistance to first-generation TKIs due to the T790M locus by irreversibly targeting the mutation site in combination with EGFR [33]. Moreover, *Her-2* and *c-MET* gene amplification are EGFR-independent mechanisms of TKI resistance. 

Significantly, the EMT program activation and transformation of small cell lung cancer pathology are also the primary mechanisms of TKI resistance [34,35]. Recently, a vast body of research has been conducted on the association between the EMT program and EGFR-TKI resistance. A previous study showed that activation of the EMT program arises in tumor tissues of erlotinib-resistant patients without other resistance mechanisms, such as T790M mutation, *c-MET*, and *Her-2* gene amplification [36]. Additional studies showed that Gefitinib-resistant PC9 and HCC827 tumor cells and Osimertinib-resistant tumor cells convert to a mesenchymal phenotype, which is manifested by the loss of cell polarity and the loosening of intercellular junctions. At the molecular level, the expression of the *epithelial markers* E-cadherin and ZO1 is downregulated, and the expression of the *mesenchymal markers* Vimentin and ZEB1 increases, illustrating the activation of the EMT process [36,37]. In addition, these studies further emphasized that excessive activation of EMT leads to acquired resistance of lung adenocarcinoma cells to EGFR-TKI [12,36,38,39]. Additionally, EMT can be a mechanism to induce EGFR-TKI resistance in lung adenocarcinoma. However, the master molecules that mediate the acquired resistance to EGFR-TKI by driving EMT occurrence remain unrevealed.

Strong evidence shows that DCLK1 is a key regulator of the EMT program in gastric cancer, colorectal cancer, pancreatic cancer, breast cancer, renal cancer, and other tumors, and silencing DCLK1 expression can inhibit the EMT process in tumor cells by down-regulating Snail and Vimentin and upregulating E-cadherin [40,41,42]. In our previous study, we found that an upregulation of DCLK1 could maintain tumor cell stemness through activation of the Wnt/β-catenin pathway and, subsequently, mediate EGFR-TKI-acquired resistance, while the application of Wnt/β-catenin pathway inhibitors only partially restored the reduced TKI sensitivity due to DCLK1 high expression, implying additional mechanism involved in TKI-acquired resistance [27]. Therefore, we speculate that the remodeling of the EMT process by DCLK1 is also an important part of its mediation of EGFR-TKI resistance. In the present study, we further disclosed that DCLK1 contributes to the activation of EMT in EGFR-TKI-resistant cells, and silencing of DCLK1 sensitizes lung carcinoma cells to EGFR-TKI by reversing the EMT process. It is suggested that DCLK1 is an effective driver gene for EMT pathogenesis and acts as a key factor to regulate EMT-related gene expression, downregulate epithelial markers, and increase mesenchymal marker expression, thereby inducing EGFR-TKI-acquired resistance. This is an essential addition to our previous studies and reinforces the role of overexpressed DCLK1 in mediating acquired resistance to EGFR-TKI. 

In addition, previous studies have reported the mechanism of DCLK1-mediated EMT genesis. In esophageal cancer-related studies, DCLK1 promotes EMT phenotype transformation through the MAPK/ERK/MMP2 pathway [43]. Furthermore, Liu et al. showed that DCLK1 can promote breast cancer metastasis by upregulating MT1-MMP, decreasing *epithelial marker* expression, and increasing Vimentin expression [44]. However, since EMT involves a group of changes in cell biological behavior due to alterations in multiple molecular pathways, it is difficult to observe EGFR-TKI sensitivity again after achieving blockade in the high DCLK1 expression state. Thus, more in-depth studies on additional mechanisms of how DCLK1 mediates EMT activation and then induces EGFR-TKI resistance are worthwhile to explore. 

In conclusion, our results indicate the role of DCLK1 in the development of EGFR-TKI resistance in lung adenocarcinoma that has undergone EMT, and targeting DCLK1 could be used as a new option for posterior-line treatment of EGFR-TKI-acquired resistance tumor. 

## Figures and Tables

**Figure 1 biomedicines-11-01490-f001:**
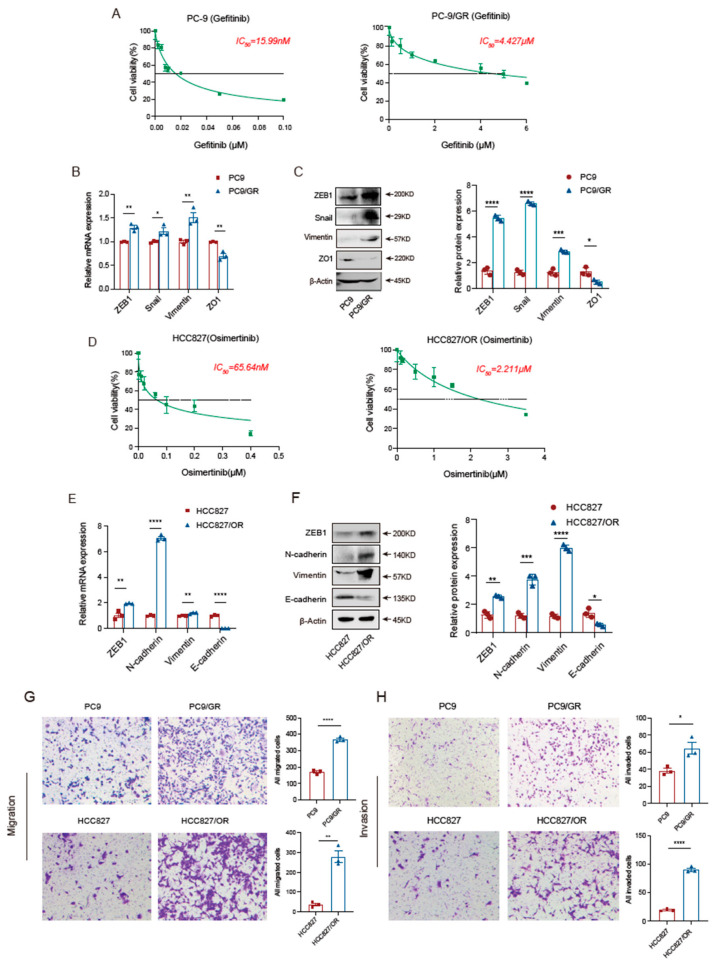
Different types of EGFR-TKI-resistant cells are universally accompanied by EMT activation. (**A**) The IC_50_ analysis of TKI-sensitive cells (PC9) and TKI-resistant cells (PC9/GR) to Gefitinib based on the CCK8 assay. (**B**,**C**) RT-PCR and Western blot results showing the EMT activation status of PC9 and PC9/GR cells. (**D**) The IC_50_ analysis of HCC827 and HCC827/OR cells to Osimertinib based on the CCK8 assay. (**E**,**F**) RT-PCR and Western blot results showing the expression levels of *epithelial markers* (E-cadherin) and *mesenchymal markers* (ZEB1, N-cadherin, and Vimentin) in Osimertinib-resistant HCC827/OR cells compared to parental cells. (**G**,**H**) The migration and invasion ability comparison of EGFR-TKI resistant cells and sensitive cells based on the Transwell assay. The data are presented as the mean ± SEM from three independent experiments performed in triplicate. * *p* < 0.05, ** *p  *<  0.01, *** *p* <  0.001, and **** *p* <  0.0001.

**Figure 2 biomedicines-11-01490-f002:**
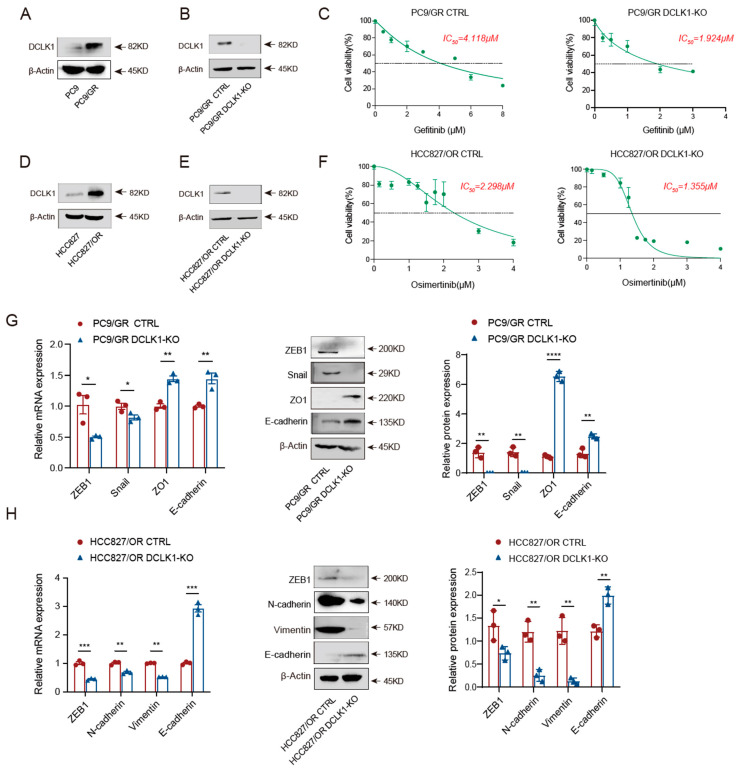
DCLK1 mediates EGFR-TKI resistance development by promoting EMT activation. (**A**) To confirm the correlation between DCLK1 expression and EGFR-TKI resistance, the expression of DCLK1 in Gefitinib-sensitive and resistant cells was detected using Western blot. (**B**) DCLK1 in Gefitinib-resistant cells (PC9/GR) was knockout using CRISPR-Cas9, and the knockout efficiency was verified using Western blot. (**C**) The IC_50_ analysis of PC9/GR CTRL and PC9/GR DCLK1-KO cells to Gefitinib based on the CCK-8 assay. (**D**) DCLK1 expression level in HCC827 and HCC827/OR cells was confirmed using Western blot. (**E**) As shown in B, the knockout of DCLK1 in HCC827/OR cells was validated using Western blot. (**F**) The IC_50_ analysis of HCC827/OR CTRL and HCC827/OR DCLK1-KO cells to Osimertinib based on the CCK8 assay. After the establishment of DCLK1-knockdown cell lines in B and E, the expression of EMT signature genes in EGFR-TKI-resistant cells compared to parental cells was detected via RT-PCR and Western blot analysis (**G**,**H**). (**I**,**J**) Comparison of the migration and invasion abilities of DCLK1-knockout and parental resistant cells using the Transwell assay. The data are presented as the mean ± SEM of three independent experiments performed in triplicate. * *p*  <  0.05, ** *p * <  0.01, *** *p* <  0.001, and **** *p* <  0.0001.

**Figure 3 biomedicines-11-01490-f003:**
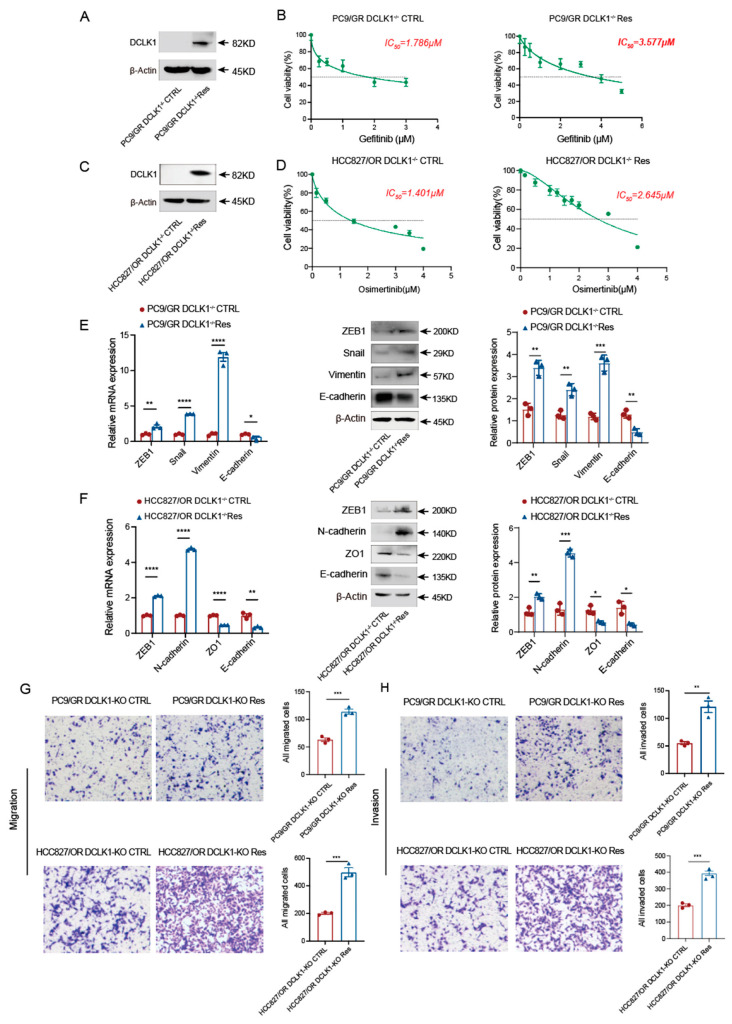
DCLK1 rescue enhances the EMT process and decreases EGFR-TKI sensitivity. (**A**) DCLK1 is re-expressed in PC9/GR knockout cells, and the expression of DCLK1 is confirmed by the Western blot. (**B**) The IC_50_ analysis of PC9/GR DCLK1^−/−^ CTRL and DCLK1-rescued PC9/GR cells (PC9/GR DCLK1^−/−^ Res) to Gefitinib using the CCK8 assay. (**C**) Western blot confirms the expression of DCLK1 in HCC827/OR knockout cells after DCLK1 rescue. (**D**) The IC_50_ analysis of HCC827/OR DCLK1^−/−^ CTRL and DCLK1-rescued HCC827/OR cells (HCC827/OR DCLK1^−/−^ Res) to Osimertinib as determined by the CCK8 assay. (**E**,**F**) EMT activation was analyzed using RT-PCR and Western blot in the DCLK1-rescued cell lines and DCLK1-knockout cell lines. (**G**,**H**) Transwell assay results of the migration and invasion abilities of DCLK1-rescued cell lines and parental DCLK1-knockout cell lines. The data are presented as the mean ± SEM of three independent experiments performed in triplicate. * *p*  <  0.05, ** *p * <  0.01, *** *p* <  0.001, and **** *p* <  0.0001.

**Figure 4 biomedicines-11-01490-f004:**
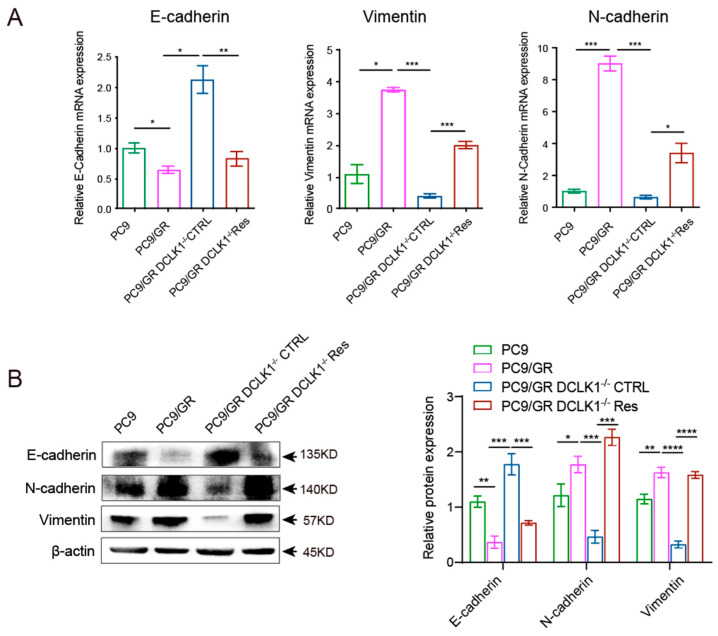
DCLK1 mediates TKI-acquired resistance by promoting EMT in vivo. (**A**) Gefitinib-sensitive PC9 cells, Gefitinib-resistant PC9/GR cells, PC9/GR DCLK1^−/−^ CTRL cells, and DCLK1-rescued PC9/GR cells (PC9/GR DCLK1^−/−^ Res) were inoculated into nude mice. The tumor tissues were collected, and the expression of *epithelial marker* (E-cadherin) and *mesenchymal markers* (Vimentin and N-cadherin) was analyzed using RT-PCR. (**B**) Tumor tissue proteins were extracted, and Western blot was performed to compare the expression of EMT-related proteins, such as E-cadherin, Vimentin, and N-cadherin, in the four groups. (**C**) Immunofluorescence analysis of EMT-related marker expression in tumor tissues. The data are presented as the mean ± SEM of three independent experiments performed in triplicate. * *p*  <  0.05, ** *p * <  0.01, *** *p* <  0.001 and **** *p* <  0.0001.

## Data Availability

Not applicable.

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
