# Peer review of "DCLK1 Drives EGFR-TKI-Acquired Resistance in Lung Adenocarcinoma by Remodeling the Epithelial–Mesenchymal Transition Status"

_biomedicines, 2023, doi:10.3390/biomedicines11051490_

Round 1
Reviewer 1 Report
The manuscript entitled “DCLK1 drives EGFR-TKI acquired resistance in lung adenocarcinoma by remodeling the epithelial-mesenchymal transition status” aims to investigate the role of the epithelial-mesenchymal transition (EMT) process in patients with lung adenocarcinoma, which is a critical mechanism that induces resistance to the epidermal growth factor receptor-tyrosine kinase inhibitor (EGFR-TKI), the treatment of choice for these patients. Furthermore, the role of doublecortin-like kinase 1 (DCLK1), a putative specific tumor cell marker, has also been studied in acquired resistance to EGFR-TKI, as it is a master regulator molecule of EMT activation.
This paper is within the scope of the journal; it is clearly explained and documented, it includes several experiments for these studies and it is in line with previous studies of this research group.
The main pros of this study is the possible applications to clinics, while the main concern is the lack of many details, mainly in the Methods section, which could hinder its reproducibility.
SPECIFIC COMMENTS
Abstract
The aim of the study is not shown.
Introduction
The last paragraph (lines 90-94) is a summary of results that should be substituted or at least accompanied by the general objective of this study, which is missing (as pointed above).
Materials and Methods
Transwell assay (lines 109-116): the parameter measured by Image J must be specified.
IC50 Determinarion (lines 130-135): specify the gradient concentrarion of Gefitinib and Osimertinib used.
Western blotting and antibodies:
- Lines 137: specify all the data related to the cells used, such as type and number of cells used, time of cells in culture.
- Line 148: specify the secondary anbibodies used and their dilution.
- Line 149: specify how the immunoreacted signals were measured and whether or not they as expressed as percentage of β-actin signal?
RNA extraction and real-time PCR:
- Line 151: specify type and number of cells, and time of cells in culture.
- Primers listed (lines 161-168) would look better in a table.
Immunohistochemistry (lines 169-178): specify the characteristic of the cells used, such as type and number of cells, and time of cells in culture, as well as the cell fixation method. Also, specify the parameter used to measure and differentiate the treatments (number of cells, darkness of DAB, etc.).
Xenograft tumor model on nude mice
- Line 184: specify the method of animal sacrifice.
- Line 184: specify the characteristic of the tissue fixed (volume, grams or number of cells) as well as the method of fixation.
- Line 185: add "as above" after ...RNA extraction.
Data and statistical analysis (lines 187-190)
Specify the statistic on each parameter described in the Results section.
Results
- All figures have a lot of data, but they are too small and therefore cannot be easily read.
- Add the statistics in the description of the result of each figure: 1 (line 211 and 345), 2 (line 231 and 362), 3 (line 247 and 376), 4 (line 264 and 385).
- In Figure 4 (lines 257-260): Immunohistochemistry should explain how the results of increased or decreased reactivity were measured.
- Lines 260-264: The last paragrph with the conclusion must go in a separate paragraph.
References
They are adequate and well written, but there are not too many recent publication, as only 35% of the references are from the last 5 years. Also, some references are missing some data, such as the pages on line 398.
GENERAL QUESTIONS
- Throughout the entire manuscript, before a parenthesis there must be a blank space. The lack of such space is observed mainly before citations.
- Between a number and the measured magnitude (such as nm, ºC, mins, etc.) there must be a blank space, throughout the manuscript.
- Line 334: delete "Figure legend".
Author Response
We thank the Reviewer for taking the time to assess the manuscript and appreciate the thorough comments and suggestions. Below you will find our point-by-point response and an account of modifications that were made to the manuscript based on the comments. We apologize for any confusion that may have arisen from our presentation of the manuscript and have done our best to increase clarity where possible.

Reviewer 2 Report
In the manuscript, “DCLK1 drives EGFR-TKI acquired resistance in lung adenocarcinoma by
remodeling the epithelial mesenchymal transition status” (biomedicines-2233898)
by Yan et al., the authors observe that two cell lines which are resistant to EGFR-kinase inhibitors show signs of a mesenchymal phenotype. Interestingly, the authors find that one single protein, DCLK1, appears to majorly influence this process. Once the authors interfere with DCLK1 expression, then also the mesenchymal phenotype is lost and can be restored, once the authors re-express it in tumour cell lines.
Based on these findings, the authors argue that a DCLK1 mediated shift from an epithelial phenotype to a mesenchymal phenotype might be causative for the resistance of the cell lines for EGFR-kinase inhibitors.
Such an assumption is novel and highly exciting. Nonetheless, additional experiments appear warranted before the presented data appear convincing. In addition, some technical issues should be addressed.
Major comments:
- if the assumption were correct that DCLK1 expression induced EMT and that EMT then causes EGFR-TKI resistance, then the pure overexpression of DCLK1 in non-resistant cell lines should also induce resistance in these cell lines. Given this were the case, then the EMT process could be blocked or the EMT process be induced in the absence of DCLK1 expression. In this way, it seems possible to dissect the EMT process from the DCLK1 mediated process and one could determine which of both processes contributes to EGFR-TKI resistance or whether one is the effect of the other.
- In its current form, Figure 4B appears not really convincing. The tumour sections should be treated for immuno-fluoresence staining and/or Western blot of the tumour should be performed. Furthermore, some sort of statistical relevant quantification should be performed. In its current form, it remains doubtful what we actually learn from such an experiment in immune deficient mice.
Minor comments:
The Material & Methods section is poorly described. The use of Gefitinib in cell culture is challenging as crystals readily form in cell culture media. Nonetheless, it is impossible to comprehend how the experiments have been performed. The authors should go through their M&M section and consider that an unexperienced researcher has to be able to repeat the experiments based on these instructions. Therefore, substantially better detailed Material & Methods section appears required.
Author Response
Thank you for your comments on our manuscript. Your comments are all valuable and very helpful for revising and improving our paper, as well as the important guiding significance to our researches. We have studied comments carefully and have made correction which we hope meet with approval.

Reviewer 3 Report
In the manuscript “DCLK1 drives EGFR-TKI acquired resistance in lung adenocarcinoma by remodeling the epithelial mesenchymal transition status”, Yan et al investigated the potential mechanism of the resistance to EGFR-TKIs and found that DCLK1 facilitates acquired resistance to EGFR-TKI in lung adenocarcinoma by induction of EMT and accelerating the migration and invasion of TKI-resistant cells. The topic is interesting, but the evidence is not convincing enough. There are several comments to help the authors improve their works.
1. The variation between experiments is very low and must be clarified that biological replicates are made. In general, the technical replicate is discouraging to perform the statistical analysis. A clear definition should be added to the legends of relevant figures. And the original raw data of the real-time PCR should be provided.
2. I seriously concerned about the western blotting. For example, 1, in the Figure 1F, 2A, and 2H, there are an additional band observed below the positive band of Actin, but other groups have not found miscellaneous band. 2, in the Figure 2H,
3. More details and data from the in vivo experiment should be provided. For example, the photo of the mice and/or xenograft tumor; the quantitative indicators of the tumor growth and mice health; Western blotting.
4. Although the authors claimed that all animal experiments were approved by the ethics committee. I have not found any detailed information and evidence to support this claim.
5. In the Figure 4B, the authors detected the expression of E-cadherin, Vimentin and N-cadherin in the mice xenograft using the Immunohistochemistry staining. Unfortunately, I have to say that all the staining signals are false positive. The positive signals suggested by the authors are belong to the background or excessive color reaction. E-cadherin and N-cadherin are mainly located in the cellular membrane. Vimentin is mainly located in cytoplasm, but not in nucleus.
Author Response
Thank you for your precious comments and advice. Those comments are all valuable and very helpful for revising and improving our paper, as well as the important guiding significance to our researches. We have studied comments carefully and have made correction which we hope meet with approval.

Round 2
Reviewer 1 Report
The authors have answered most of the questions in detail, the majority in the methodology, and therefore the paper has improved and increased clarity.
However, there are still a few questions left:
- Table 1 is shown in the supplementary data but should be mentioned in the text (line 183).
- The authors have changed the method for immunostaining, and beautiful images are observed, however, in the description of the method of this technique, the information related to the incubation time of the antibodies, or the temperature is missing (lines 201-211).
- The authors reply to a question regarding immunoreactive signals with the following sentence, which should somehow be included in the text (line 171-on): "All immunoreacted signals were measured using Image J software, with control signals normalized and presented as a percentage of the control group. We have supplemented the grayscale analysis results in the text as follows"
- The figures contain a lot of data, but the panels they contain are so small that they are difficult to read, and it is difficult to differentiate the data, as I mentioned in the 1st revision. The authors mention that they have increased the size, but they have also added new panels to some of the figures and are therefore now even smaller than in the 1st version of the manuscript.
Minor questions
- The symbol (#) should appear for the corresponding authors, instead of the symbol (*) (line 7).
- A blank space must appear before the units of measurement, such as mL, min, h, etc, as well before periods (.), throughout the manuscript.
Author Response
Thank you for reviewing our manuscript and offering valuable advice. Your comments are very important and have greatly helped us to revise the manuscript. We have revised the manuscript point by point according to your suggestions. All changes are highlighted in red, and text without changes compared to 1st version is highlighted in blue.
Reviewer 2 Report
The authors have very nicely responded to my comments and experimentally addressed all those aspects I was requesting. I feel these additional experiments have substantially improved the quality of the manuscript.
In addition, the authors have diligently discussed my criticism with regard to the underlying mechanism that may lead to the observed effects. Currently, the data do not reveal whether the observed effect is mediated by the cellular process of EMT (as it is currently published in several publications) or via the described DCLK1 pathway. In their response to my criticism, the authors very nicely discussed this issue. Nonetheless, I would much appreciate if the authors could do so in the Discussion section of the manuscript, as well.
I feel it would be important to put their finding in the context of other developments in the field - and in specific to those recent publications, which showed alternative mechanisms of how EMT can contribute to EGFR-TKI resistance.
Author Response
Comments and Suggestions for Authors
The authors have very nicely responded to my comments and experimentally addressed all those aspects I was requesting. I feel these additional experiments have substantially improved the quality of the manuscript.
In addition, the authors have diligently discussed my criticism with regard to the underlying mechanism that may lead to the observed effects. Currently, the data do not veal whether the observed effect is mediated by the cellular process of EMT (as it is currently published in several publications) or via the described DCLK1 pathway. In their response to my criticism, the authors very nicely discussed this issue. Nonetheless, I would much appreciate if the authors could do so in the Discussion section of the manuscript, as well.
Response: Thank you very much for your valuable comments. We are very pleased that our response can address your concerns. In accordance with your suggestions, we have compiled your responses into the discussion section and revised the text details to ensure the quality of the article. In addition, the references cited in the discussion section have also been updated. A total of 5 references have been added. All modifications above are marked in red. We thank you again for your review of our manuscript. The minor version of the manuscript is attached.

Reviewer 3 Report
The authors' submission lacks updated figures, which is crucial for me to evaluate their work thoroughly. Without a complete manuscript, it is difficult for me to assess the validity and significance of their findings.
Author Response
Sincerely thank you for your valuable comments. Your comments are extremely vital to us to revise the article and improve the accuracy and clarity of our article.
Each comment was directly addressed regarding the modified manuscript with changes highlighted in red. We have checked the manuscript carefully. Here, in response to the question you raised in the text, we have changed all wording that is not rigorous and thanks for your attention. The new version of the manuscript is attached.

Round 3
Reviewer 3 Report
Thank you for the author's reply. I appreciate that most of my concerns have been addressed, and the quality of this manuscript has significantly improved. However, I would like to note that there are still several minor mistakes and errors that require attention before considering publication. The errors I listed may be inaccuracies. The authors should review their manuscript carefully and make any necessary modifications to ensure accuracy and clarity.
In the abstract section,
"Epidermal growth factor receptor-tyrosine kinase inhibitor (EGFR-TKI) is the first-line treatment" should be "EGFR-TKI is a first-line treatment."
"acquired resistance to EGFR-TKIs remains unavoidable" should be "acquired resistance to EGFR-TKIs remains a problem."
"development of Epithelial-mesenchymal transition (EMT) is one of the critical mechanisms that induce acquired resistance to TKI" should be "development of Epithelial-mesenchymal transition (EMT) is a critical mechanism that induces acquired resistance to TKIs."
"Reverse EGFR-TKI acquired resistance" should be "Reversing acquired resistance to EGFR-TKIs."
"aimed to explore the role of Double-cortin-like kinase 1 (DCLK1) as an EMT driver gene in acquired resistance of lung adenocarcinoma to EGFR-TKI" should be "aimed to explore the role of Doublecortin-like kinase 1 (DCLK1) as an EMT driver gene in the acquired resistance of lung adenocarcinoma to EGFR-TKI."
"The expression level of EMT-related genes in PC9 and HCC827 cell was detected" should be "The expression level of EMT-related genes in PC9 and HCC827 cells were detected."
"For in vivo experiments, PC9 cells were subcutaneously (s.c) injected into BALB/c Nude mice for tumor formation, upon harvestation, tumor tissues were retained" should be "For in vivo experiments, PC9 cells were subcutaneously injected into BALB/c Nude mice to form tumors. Upon harvesting, tumor tissues were retained."
"Gefitinib-resistant PC9 (PC9/GR) and Osimertinib-resistant HCC827 (HCC827/OR) cells showed remarkable activation of EMT and enhanced ability of migration and invasion compared to TKI-sensitive cells" should be "Gefitinib-resistant PC9 (PC9/GR) and Osimertinib-resistant HCC827 (HCC827/OR) cells showed remarkable activation of EMT and enhanced migration and invasion abilities compared to TKI-sensitive cells."
"DCLK1 expression was increased markedly in EGFR-TKI resistant lung adenocarcinoma cells" should be "DCLK1 expression was markedly increased in EGFR-TKI-resistant lung adenocarcinoma cells."
"Targeted knockout of DCLK1 effectively reversed the EMT phenotype in TKI-resistent cells, with improved EGFR-TKI sensitivity" should be "Targeted knockout of DCLK1 effectively reversed the EMT phenotype in TKI-resistant cells and improved EGFR-TKI sensitivity."
" DCLK1 facilitates acquired resistance to EGFR-TKI in lung adenocarcinoma by inducting of EMT and accelerating the migration and invasion of TKI-resistant cells" should be " DCLK1 facilitates acquired resistance to EGFR-TKI in lung adenocarcinoma by inducing EMT and accelerating the migration and invasion abilities of TKI-resistant cells."
In the introduction section,
"Non-small cell lung cancer (NSCLC), the most common histological subtype of lung cancer, accounts for about 85% of lung cancer worldwide, and lung adenocarcinoma is the one that needs urgent concern [1,2]." - This sentence is a bit lengthy and could be revised for clarity. One possible way to rephrase it is: " Lung cancer is a pressing global health concern, with non-small cell lung cancer (NSCLC) being the most prevalent subtype, representing approximately 85% of all cases. Within the spectrum of NSCLC subtypes, lung adenocarcinoma warrants urgent attention due to its high incidence rates. "
"In Asian population, 50-60% of lung adenocarcinoma patients have EGFR-sensitive mutations (exon 19 deletion mutation, exon 21 L858R point mutation) [3]." - "Asian population" should be "the Asian population." Also, "exon 19 deletion mutation, exon 21 L858R point mutation" should be "the exon 19 deletion mutation and the exon 21 L858R point mutation" or "exon 19 deletion and exon 21 L858R mutations."
"Recent studies have shown that excessive activation of EMT will lead to acquired resistance of lung adenocarcinoma cells to EGFR-TKI [12]." - "Will lead" should be changed to "can lead" for greater accuracy.
"In addition, many researches suggest that the EMT process occurs independently or concomitantly with other resistance mechanisms, such as EGFR-T790M locus mutations or MET gene amplification, in patients with acquired resistance to EGFR-TKI [13]." - "Many researches" should be "Many researchers." Also, "EGFR-T790M locus mutations" should be "mutations in the EGFR-T790M locus.”
"Doublecortin-like kinase 1 (DCLK1) is a microtubule-associated protein initially identified in the nervous system and has been shown involved in neurogenesis and microtubule migration [14,15]." - "has been shown involved" should be changed to "has been shown to be involved."
"Crystallgraphy study revealed that DCLK1 contains multiple functional structure domains." - "Crystallgraphy" should be "Crystallography."
"In addition, DCLK1 is a crutial player of EMT activation through regulating the expression of EMT-associated genes, including the upregulation of mesenchymal markers Snail, N-cadherin, and ZEB1, and downregulation of epithelial markers ZO1 and E-cadherin [22,23]." - "Crutial" should be "crucial." Also, "upregulation of mesenchymal markers Snail, N-cadherin, and ZEB1, and downregulation of epithelial markers ZO1 and E-cadherin" could be rewritten for greater clarity. The revised sentence would be: "In addition, DCLK1 plays a crucial role in activating EMT by regulating the expression of EMT-associated genes. It upregulates mesenchymal markers such as Snail, N-cadherin, and ZEB1, while downregulating epithelial markers such as ZO1 and E-cadherin [22,23]."
In the Materials and Methods section,
"EGFR-TKI-sensitive lung adenocarcinoma cells PC9 and HCC827 were obtained from Shanghai FuHeng biological Co., Ltd and American Type Culture Collection (ATCC), respectively.”, I have not found the company you listed here, but I found Shanghai fuheng biotechnology co. ltd?
"Both are epithelial adherent cells of lung adenocarcinoma with EGFR 19 exon deletion mutation." --> "Both are epithelial adherent cells of lung adenocarcinoma with EGFR exon 19 deletion mutation."
"PC9 Gefitinib-resistant cells were purchased from Shanghai FuHeng biological Co., Ltd, defined as PC9/GR, with a Gefitinib resistance concentration of 2.2μM." --> "PC9 Gefitinib-resistant cells, defined as PC9/GR, were purchased from Shanghai FuHeng Biological Technology Co., Ltd and have a Gefitinib resistance concentration of 2.2μM."
"HCC827 Osimertinib-resistant cell line was provided by Beijing Chest Hospital, defined as HCC827/OR, with an Osimertinib resistance concentration of 0.16μM." --> "The HCC827 Osimertinib-resistant cell line, defined as HCC827/OR, was provided by Beijing Chest Hospital and has an Osimertinib resistance concentration of 0.16μM."
"PC9 and PC9/GR cells were cultured with Dulbecco’s minimal essential medium (DMEM), while HCC827 and HCC827/OR cells were cultured with RPMI-1640 medium." --> "PC9 and PC9/GR cells were cultured in Dulbecco’s modified Eagle’s medium (DMEM), while HCC827 and HCC827/OR cells were cultured in RPMI-1640 medium."
"10% Fetal Bovine Serum (FBS) and 1% penicillin streptomycin (P/S) were supplemented to the basic medium and configured as a complete medium." --> "The basic medium was supplemented with 10% fetal bovine serum (FBS) and 1% penicillin-streptomycin (P/S) to create a complete medium."
"The incubator conditions are set as 37 ℃, 5% CO2." --> "The cells were incubated under standard conditions of 37°C and 5% CO2."
"supplemented with 1% protease inhibitor and 1% Phenylmethanesulfonyl fluoride (PMSF)" - it would be clearer to use the passive voice here and say "1% protease inhibitor and 1% Phenylmethanesulfonyl fluoride (PMSF) were added to the RIPA lysis buffer."
"Whole cells were lysed with RIPA buffer on ice and protein quantification was performed using the BCA Protein Assay Reagent" - it would be clearer to use parallelism here and say "Whole cells were lysed with RIPA buffer on ice, and protein quantification was performed using the BCA Protein Assay Reagent."
"The same amounts of denatured proteins were separated by electrophoresis in 10% SDS-PAGE and transferred to PVDF membranes activated by methanol" - it would be clearer to use the passive voice here and say "The denatured proteins were separated by electrophoresis in 10% SDS-PAGE, and then transferred to PVDF membranes activated by methanol in equal amounts."
"the PVDF membranes were blocked in 8% nonfat milk, ready for the incubation of primary antibody for 4 ℃ over-night" - it would be clearer to say "the PVDF membranes were blocked with 8% nonfat milk and incubated with primary antibodies overnight at 4℃."
"After the PVDF membrane was washed with 1×TBST buffer 3 times, the specific HRP-conjugated secondary antibodies" - it would be clearer to say "After washing the PVDF membrane with 1×TBST buffer three times, specific HRP-conjugated secondary antibodies were incubated for 60 minutes at room temperature."
"then the immunoreacted signals were visualized by ECL reagents" - it would be clearer to say "The immunoreacted signals were visualized using ECL reagents."
Lines 287-300, "rescued" should be changed to "rescue" to maintain tense consistency; "verified" should be followed by "the" to make the sentence grammatically correct; the word "versus" should be replaced with "in" to convey the intended meaning; the phrase "nearly 2-fold higher" should be changed to "almost two times higher" to improve clarity; "recapitulation" should be replaced with "reintroduction" for better accuracy; the word "enhance" should be changed to "enhances" to match the subject-verb agreement; "suggest" should be changed to "suggests" for subject-verb agreement.
Lines 317-331, "tumor tissues were harvested" should be changed to "we harvested tumor tissues" to improve sentence structure."RT-PCR" should be followed by "results showed" for better clarity."while" should be replaced with "whereas" for better sentence structure."re-expressed" should be hyphenated as "reexpressed" for correct spelling."E-cadherin" should be capitalized for consistency with other mentions in the text.the phrase "In contrast" should be moved to the beginning of the sentence for improved coherence.
In the discussion section,
"EGFR sensitive mutations" - "Sensitive" should be changed to "sensitizing" to make it "EGFR-sensitizing mutations".
"since that acquired resistance" - "That" should be changed to "as" to make it "since acquired resistance".
"Among them" - This phrase is not connected to the previous sentence and should be rephrased or connected using conjunctions like "However" or "In addition".
"first-generation TKI" and "third-generation TKI" - These should be written as "first-generation TKIs" and "third-generation TKIs", respectively.
"irreversibly in combination with EGFR" - This phrase seems incomplete and should be rephrased or expanded.
"vast body of research regarded" - "Regarded" should be changed to "regarding" to make it "vast body of research regarding".
"PC9, HCC827 Gefitinib-resistant" - "Gefitinib-resistant" should be hyphenated to make it "Gefitinib-resistant PC9 and HCC827".
"tumor" - There is a missing word after "tumor". It should be "tumor cell".
Author Response
Sincerely thank you for your valuable comments. Your comments are extremely vital to us to revise the article and improve the accuracy and clarity of our article.
Each comment was directly addressed regarding the modified manuscript with changes highlighted in red. We have checked the manuscript carefully. Here, in response to the question you raised in the text, we have changed all wording that is not rigorous and thanks for your attention. Please see the attachment.
